# Spreading potential in disease relevant networks: Predicting centralities in rural Northeast Madagascar

Camille M. M. DeSisto[1,2�'s], Raquel A. Binder[3�s*], Kayla Kauffman[4], Tyler M. Barrett[5], Michelle Pender[6], Randall A. Kramer[1,6], Voahangy Soarimalala[7,8], Jean Yves Rabezara[7], Prisca Rahary[7], James Moody[9], Charles L. Nunn[5,6]

1 Nicholas School of the Environment, Duke University, Durham, North Carolina, United States of America, 2 Sustainability Institute, Rice University, Houston, Texas, United States of America, 3 Department of Medicine, University of Massachusetts Chan Medical School, Worcester, Massachusetts, United States of America, 4 Department of Ecology, Evolution and Marine Biology, University of California, Santa Barbara, California, United States of America, 5 Department of Evolutionary Anthropology, Duke University, Durham, North Carolina, United States of America, 6 Duke Global Health Institute, Duke University, Durham, North Carolina, United States of America, 7 Association Vahatra, Antananarivo, Madagascar, 8 Institut des Sciences et Techniques de l'Environnement, Université de Fianarantsoa, Fianarantsoa, Madagascar, 9 Department of Sociology, Duke University, Durham, North Carolina, United States of America

☹ These authors contributed equally to this work.
* raquel.binder@umassmed.edu

## Abstract

Heterogeneity in contact patterns can have marked effects on disease transmission, including through superspreading where few individuals drive most infections. Networks based on different types of human-human contacts quantify individuals' centrality, which can be used to identify individuals or sub-populations who are at increased risk of spreading disease. By understanding the predictors of centrality, high-risk individuals and sub-populations can be targeted to improve public health intervention strategies, even when detailed network data are unavailable. This study inferred transmission potential networks representing different pathogen transmission pathways among people living in rural villages of northeast Madagascar. We constructed four network types: social, close contact, household proximity, and environmental overlap using survey data and global positioning system (GPS) trackers. We then investigated how sociodemographic and anthropometric variables predicted different types of network centralities using multiple mixed effects linear models. Gender and wealth based on household material quality tended to be the most important sociodemographic predictors of centrality, but centrality outcomes varied by network type and had wide confidence intervals. Men tended to be more central to their environmental overlap network than women. Further, wealth based on household materials was an important, positive predictor of close contact network centrality. Gender and wealth were associated with centrality in transmission-potential networks but

**Data availability statement:** We are unable to share even a minimal anonymized data set because a clause in the consent form stated that individual answers would not be shared publicly, which has been interpreted by the IRB as not sharing any individual level survey data. Likewise, GPS data, which cannot be anonymized, cannot be shared. The Duke University Campus IRB advised on the restriction of individual level data and can be contacted through campusirb@duke.edu. The R code for the analysis is available on Github: https://github.com/MadagascarEEID/NetworkCentrality.

**Funding:** This work was supported by the joint NIH-NSF-NIFA Ecology and Evolution of Infectious Disease program (R01-TW011493–01 to CLN and JM), the NSF Predictive Intelligence for Pandemic Preparedness program (BCS-2200047 to CLN and JM), and the National Center for Advancing Translational Sciences (KL2-TR001455 to RAB). The funders had no role in study design, data collection and analysis, decision to publish, or preparation of the manuscript.

**Competing interests:** The authors have declared that no competing interests exist.

varied in their importance across different network types. Our study results suggest that targeted intervention efforts focused on diseases that are transmitted through shared environments (i.e., parasites shared through soil or water) or direct contact (i.e., respiratory infections) in similar agricultural settings should consider gender- and wealth-associated differences in contact patterns.

## Introduction

Infectious diseases spread through contact networks, where nodes (i.e., people) and their edges (i.e., connections between people) represent epidemiologically relevant contacts for disease transmission (e.g., sexual interactions for sexually transmitted diseases). The number of edges connecting one node to others, the intensity and duration of those contacts, and the location of a node in a network determine the node's relevance for disease transmission [1,2]. These networks represent possible pathways for disease spread and are known as *transmission potential networks*. Examples include close contact networks that represent potential pathways for the spread of respiratory viruses and environmental overlap networks that represent potential pathways for the spread of soil-transmitted helminths. The importance of a node in these networks is often quantified using centrality metrics [1,2].

Identifying individuals who are central to their network, and therefore likely to disproportionately drive disease transmission, is essential for the design and implementation of effective infectious disease control interventions [3]. Indeed, a small percentage of individuals within any population are believed to be responsible for most secondary infections (i.e., superspreaders; the 20/80 rule) [4–7]. Superspreading has been documented for SARS-CoV-1, SARS-CoV-2, MERS-CoV, measles, and Ebola, among others [6–10]. For example, in Singapore, from 201 probable SARS patients, five individuals infected ten or more secondary contacts each, while 81% had no evidence of infecting others [5,11,12].

While having detailed information on network position would be ideal for identifying pathogen transmission potential, collecting contact data to build transmission potential networks is expensive and time-consuming. Alternative low-cost and effective means to identify individuals who are central to their networks prior to an outbreak is therefore critical, particularly in limited-resource settings where diagnostics and vaccines are scarce and the health care infrastructure is less developed [13–18].

Network centrality is shaped by heterogeneity in contact patterns, which in turn is shaped by a mix of individual, physical, behavioral, cultural, and environmental characteristics. Previous studies have considered core demographic variables, such as age and gender, to predict social contact rates [19,20]. For example, Prem *et al.* [20] estimated age-and-location-specific contact patterns at home, work, school and other locations in 152 countries. They found that contacts were highly assortative by age across all countries, and that there were more inter-generational contacts in Asian countries than in other settings. Fewer studies have investigated the influence of occupation, socioeconomic status, movement patterns, and anthropometric

characteristics on contact heterogeneity and the relevance of these relationships for modeling infectious disease transmission [21].

Previous research has shown that targeting individuals who are highly connected in human contact networks based on demographic predictors of centrality can reduce infection burden more effectively than random interventions, even when demographic predictors of centrality are weak [19]. Identifying disease transmission network heterogeneities, particularly in rural, low-resource contexts, is therefore critical for decreasing infection burden.

Here, we investigated how sociodemographic and anthropometric variables predicted network centrality in three rural villages in northeast Madagascar. We analyzed the following predictors of network heterogeneities: gender, age, body mass index (BMI), education, and several socio-economic indices (owned durable goods, household size, house construction materials, land size, and number of livestock owned). These variables are relevant to infectious disease transmission in rural low-resource settings because they shape close-contact interactions and environmental overlap through social interactions and physical activities [14,19,22]. We investigated the predictors of centrality across four networks, representing multiple transmission pathways – social, close contact, household, and environmental networks. We assessed a range of centrality measures to capture different aspects of node importance, including strength, PageRank, eigenvector, and betweenness.

## Methods

### Data collection

Sociodemographic survey and GPS movement data were collected from adult study participants (n = 1,220) in three villages (Village A, S, and M) in the Sambava, Andapa, Vohemar, Antalaha (SAVA) region of Madagascar near Marojejy National Park from October 3, 2019, to August 4, 2022. The study was approved by the Duke University IRB (protocol no. 2019–0560). The Malagasy Ethics Committee for Biomedical Research within the Ministry of Public Health reviewed the study documents and survey, concluding it did not require formal IRB review. Additional information regarding the ethical, cultural, and scientific considerations specific to inclusivity in global research is included in the Supporting Information, see S1 Checklist.

### Population & sampling approach

The study covered three villages along the boundary of Marojejy National Park in the SAVA region of northeastern Madagascar (see map in S1 Fig). Villages remained unnamed to protect the participants in the small villages (i.e., participants may be identifiable based on age, sex, and household characteristics due to the small village sizes).

We drew upon existing relationships with the community and village leadership from Village M and selected nearby Villages A and S as additional data collection sites to conduct this study [23–27]. We engaged the communities through group meetings and direct consultations with village leaders. All three villages are forest frontier communities adjacent to Marojejy National Park and village residents are primarily farmers who grow crops such as rice and vanilla to support their livelihoods [25]. Located near the main entrance to Marojejy National Park, Village M is the primary access point for tourists. Village A is the most rural and composed of two communities that are less than 1 km apart, and have shared familial ties and farmland. We therefore considered the two communities one village (Village A). Village S is also adjacent to the national park yet lies within the broader Andapa Basin that is a major rice-growing area.

We surveyed each village for 3–6 weeks over multiple seasons. Season 1 was hot and dry, Season 2 was warm and wet, and Season 3 was cool and wet (transitional). Villages S and A were surveyed in all three seasons. Village M was surveyed in Seasons 1 and 2. Malagasy team members, fluent in the local dialect, translated the study questions to Malagasy, ensuring the intent and meaning of the questions were maintained. The survey was pre-tested with a small group of community members in the villages to ensure clarity and receive constructive feedback for fine-tuning the survey

instruments. To support informed participation, Malagasy researchers, together with a trusted community representative, provided detailed explanations of the study and created a setting where individuals could freely raise concerns or opt out.

Adult study participants (18 + years of age) were enrolled through a snowball sampling method [28]. Our team started the enrollment with a small group of landowners and their households, while other adult individuals in the villages became eligible for enrollment once they were named as someone with whom enrolled participants spend their free time, exchange farmwork, or exchange food (see Table 1). While the snowball method will naturally miss truly isolated people, (a) our substantive engagement with the population suggested that this was not a concern because individuals had large extended family and coworking networks, (b) from a disease transmission standpoint, truly isolated nodes were not a concern, and (c) prior research on this topic revealed that, for networks similar to ours, degree measures maintained a high correlation (≈0.9) even with >70% of missing data [29–31]. To assess whether centrality declined over time and its potential impact on our findings, we included a sensitivity analysis with sampling date as a predictor (see Supplemental Material).

After informed consent was obtained from participants (oral consent for survey and written consent for GPS), we collected individual data (gender, age, BMI, education, owned durable goods at household level, and GPS movement data) and household data (household size, house construction material, land size, number of livestock owned, and GPS location of home).

Weight (kg) and height (meters) were measured by our team and used to calculate BMI [32]. We considered the social/cultural variable gender rather than the biological variable sex. Education was based on self-reported level of formal education, categorized as no formal schooling, completion of primary school, secondary school, or higher education, and then coded as an ordinal variable. Owned durable goods were calculated as the sum of the following items reported by each participant: cell phone, television, bicycle, refrigerator, motorcycle, computer, and electricity generator. The house material index was calculated as the sum of codes for different materials used in the construction of household walls (bamboo/ raffia palm/ ravenala palm/ mud/ compacted earth = 0, wood planks = 1, bricks = 2, metal sheets = 3, cement = 4), roofs (bamboo/thatch = 0, metal sheets = 1, cement = 2), and floors (dirt/ bamboo/ raffia palm/ ravenala palm = 0, wood planks = 1, cement = 2), with a higher number (i.e., index) representing a more durable home structure that requires greater financial means and engagement with local markets to procure goods and construct the house. Livestock ownership was quantified as the self-reported sum of cows, goats, and pigs owned. All variables except for gender were treated as continuous and standardized.

To reduce the dimensionality of our wealth data, we conducted a principal components analysis (PCA) (see S1 Text and S2 Fig). The variance in wealth was distributed across multiple components, and principal component 1 only explained

**Table 1. Network measurement approaches. For detailed methodology of network construction, including edge weights, see Kauffman et al. 2022.**

| Network | Method | Outcome Measure | Edge Weight (for each pair) |
|---|---|---|---|
| Social | Social network surveys in which individuals named up to five other people who they: (i) meet in their free time, (ii) receive farming help from, (iii) give farming help to, (iv) receive urgently needed food from, and (v) give urgently needed food to. | Self-reported social ties/interactions | Sum of reported social interactions (1–10) |
| Close Contact | GPS-based contact, volume intersections, naming network ties, and demographic characteristics (gender and age similarity) | Individual proximity | Estimated probability of interaction (0–1) |
| Household | The inverse of distance between participant homes based on GPS locations. | Household proximity | Inverse of the geographic distance between houses (0.00-0.93) |
| Environmental | Time spent in shared locations *outside of villages* based on landcover classifications and utilization distributions. | Environmental overlap (i.e., co-use of space) | Product of the proportion of utilization distribution, summed across grid cells (0.00-0.25) |

38.4% of the variation. Since our wealth data were multidimensional (i.e., widely distributed across the principal components, see S2a Fig), we analyzed distinct dimensions of wealth separately. We also avoided categorizing households into wealth quintiles to retain the continuous nature of the underlying variables, thereby preserving the ability to detect fine-scale variation in wealth among individuals.

## GPS data collection

Trained Malagasy assistants explained the geospatial aspect of the study alongside the survey component. To ensure participant awareness, we explained how the GPS trackers worked, presented a map of the area with tracker trajectories, and stated "people's movements will be tracked" by these devices. Participants were compensated with pre-paid cell phone cards worth approximately $USD 0.75 for each week of participation. The amount was determined in consultation with the IRB and local team to safeguard against undue influence. Participants had the option to receive similar compensation by participating in the survey and other aspects of the study without wearing the GPS. We did not require GPS checks or participants to finish the survey for compensation. GPS consent was obtained in writing, separate from the survey consent, and included information on the technology and how the data would be secured and used.

We used GPS data loggers (iGot-U 120; Mobile Action, New Taipei City, Taiwan) to record participant movement. Participants had the option to wear a GPS tracker one week at a time and up to six weeks in total. Following each week, GPS units were collected and replaced with fully charged devices for those who wished to continue in the study. The participants were asked to always wear the trackers during the day. Devices recorded participant location every 3 minutes.

The GPS-based tracking data were processed as described in Kauffman et al. 2022 [14]. Briefly, we first removed erroneous points, the distribution day, and days that participants did not wear the device, determined by self-reporting and individuals' daily minimum convex polygon and trajectory. Next, we determined the study area around each village using a minimum convex polygon based on recorded locations. We limited our analysis to the active period (i.e., daytime hours, 06:00–19:59 GMT + 3). Because individuals did not all wear GPS units simultaneously, GPS trajectories were pooled at the weekly level to generate individual utilization distributions (UDs), which allowed estimation of spatial overlap even for dyads that were not tracked at the same time. Individuals' daytime utilization distributions, which are the probability density estimates of individuals occurring at a given location [33], were calculated using a dynamic Brownian Bridge Movement Model [34]. For additional details, see Kauffman et al. 2022 [14].

## Network types

We constructed four types of networks that represented different types of potential contacts. (i) *Social networks* were based on surveys where participants named others with whom they spend free time and/or exchange food and/or co-farmwork [35]. Social networks captured longer-term social relationships and pathogen transmission potential through direct contact and shared food, water, and space. (ii) *Close contact networks* were based on GPS data, social network ties, and demographic characteristics. These networks captured individual proximity and reflected potential for contact-based transmission. (iii) *Household networks* were based on household distance within each village and captured potential pathogen transmission through direct contact, shared environments around the house, water, and vectors. (iv) *Environmental networks* were based on GPS-derived environmental overlap outside of the village and captured potential environmental transmission based on shared land use, including parasite transmission in soil or water or vector-borne diseases. All four network types were constructed for each of the three villages, resulting in a total of 12 networks.

More details on network construction are provided in Table 1 and in Kauffman *et al.* [14]. Spatial overlap was based on individual GPS tracker data points, along with additional data to predict close contact when individuals did not wear GPS trackers at the same time. Environmental overlap for the environmental networks was defined by the proportion of each individual's 95% UD falling within a 30x30m moving window of 10x10m classified Sentinel II imagery. The overlap weight

between two individuals equaled the sum of the products of their UD proportions across cells, providing a continuous measure of shared environmental exposure [14]. We limited the environmental network to cells outside of villages (i.e., in forest, shrubby regrowth, agricultural fields, water, or bare ground).

## Network statistics

All analyses were conducted using R Version 4.3.1 [36]. For each network, we used the R package igraph [37] to calculate four centrality metrics: strength, PageRank, eigenvector, and betweenness. Strength centrality is the sum of edge weights of each node and is a centrality measure that has been applied in other infectious disease studies [2,38]. To calculate strength, the edges were weighted to reflect different amounts of contact and then summed at each node. PageRank centrality accounts for the importance of a node's neighbors and incorporates the number of edges and edge weights as it simulates an expected steady-state population of random walkers on a network [39]. It mirrors possible movement of a pathogen or information through a network and is therefore a useful centrality measure when investigating disease transmission [40]. Eigenvector centrality (also known as power centrality) reflects the degree to which a node is connected to other highly connected nodes, and thus also captures connectivity of nodes, making it well suited for studies of infectious disease transmission [2,38]. Here, eigenvector centrality was the value of the first eigenvector of the network adjacency matrix. Betweenness centrality quantifies the importance of a focal node by calculating the number of shortest paths between all other pairs of nodes that pass through the focal node [41]. Thus, a node with high betweenness centrality is better at connecting other nodes in the network, making betweenness relevant to studies of infectious disease transmission [2,13,38].

### Predictors of centrality

We assessed associations between examined variables (sociodemographic and anthropometric characteristics) and centrality using multiple mixed effects linear regression. Standardized centrality was the dependent variable, and the assessed characteristics were independent (predictor) variables. In addition to sociodemographic and anthropometric characteristics, we controlled for type of centrality metric (strength, PageRank, eigenvector, and betweenness), village, and sampling season as independent variables. Individual (i.e., node ID) was included as a random effect. Models were fit in the package lme4 [42]. We conducted model averaging with shrinkage using the package MuMIn [43]. The estimates listed in the results and supplemental tables are the normalized estimated effect size of the variable on log centrality. We quantified the importance of each predictor variable based on the sum of Akaike information criterion (AIC) weights across the set of models used in model averaging (range 0–1; 0 being the least and 1 being the most important) [43]. Importance represents the probability that a given variable is included in the model, out of the comprehensive suite of models used in the model averaging. The highest importance score of 1 means that, across our model set of all candidate models used in model averaging, that variable is extremely likely (estimated 100%) to be included in the models that best describe the predictors of centrality for that specific network. For example, a gender importance score of 1 in the environmental network indicated that, across our model set of all candidate models used in model averaging, the gender variable was very likely to be included in the model that best described the variation in centralities in the environmental network. We evaluated models for heteroskedasticity and multicollinearity and found them to be within reasonable ranges. We also conducted a sensitivity analysis of our snowball sampling methodology by fitting models with sampling date index (relative order of sampling date across the study period). These results are provided in S3 Fig and S4 Fig.

To examine specific aspects of node relevance that were captured by different centrality measures, we also built separate statistical models for each centrality metric (n = 4), network type (n = 4), village (n = 3), and sampling season (n = 3 for villages A and S, n = 1 for village M) combination, for a total of 124 models. Again, we conducted model averaging with shrinkage. Fixed-effects models were fit across villages and seasons and model effect sizes were pooled in a

meta-analysis approach using the package metafor [44]. This resulted in 16 models that were subset by centrality metric and network type. Results for these models are found in S5 Fig and S1 Table.

## Results

### Study participants

Demographic, health, socioeconomic, household, and agricultural data from 1,220 study participants were collected from Village A (n = 563), Village S (n = 408), and Village M (n = 249) from 2019 to 2022 (Table 2). Originally, 1,297 people were interviewed for the study and 77 participants were omitted due to missing data (n = 1,220).

### Transmission potential networks

To build intuition about the differences in structure across network types and settings, we visualized each network (Fig 1). Nodes were located within the plots based on a force-directed layout algorithm (Fruchterman Reingold [45]), which generally places nodes tied together near each other in space. Because physical distance-based networks are maximally dense, we backboned the network to retain only the strongest ties for each node. In cases where this procedure created disconnected graphs, the placement of isolates/small components was arbitrary. However, there were no disconnected nodes in the networks used for statistical analysis.

The close contact and household network structures resembled each other, highlighting the correlation between individual and household proximity. The social network differed in structure and appeared more evenly connected, indicating that the social ties were less fragmented and overcame physical distances between individuals.

### Predictors of network centrality

Of the sociodemographic predictors, gender and wealth (measured in terms of house material index; higher index values indicating higher quality house materials) were the most important sociodemographic predictors of centrality, but results varied by network type (Fig 2). For full model results, see S2 Table.

For the social network, gender was the most important individual or household level centrality predictor (importance = 0.51, estimate = 0.06, estimate 95% CI = -0.07-0.18), and men tended to have higher centrality than women (Fig 2a). Livestock ownership also predicted centrality (importance = 0.36, estimate = 0.02, estimate 95% CI = -0.04-0.08), with number of livestock owned being positively associating with centrality (Fig 2a). However, the confidence intervals for both gender and livestock ownership on social network centrality overlapped zero. Other variables were not important predictors of social network centrality.

For the close contact network, the house material index was the most important individual or household level predictor of centrality (importance = 0.96, estimate = 0.08, estimate 95% CI = 0.03-0.14). Age (importance = 0.32, estimate = 0.02, estimate 95% CI = -0.04-0.08) and gender (importance = 0.20, estimate = -0.02, estimate 95% CI = -0.09-0.06) were weak predictors of close contact network centrality, and other variables were not important (Fig 2b). Overall, participants who were wealthier (house made of higher quality materials), older, and men tended to be more central to their close contact networks (Fig 2b). None of the assessed individual or household level variables were important predictors of household distance network centrality (Fig 2c).

For the environmental network, gender (importance = 1, estimate = 0.39, estimate 95% CI = 0.27-0.50) was the most important individual or household level predictor of centrality, with men tending to be more central than women (Fig 2d). The house material index (importance = 0.29, estimate = 0.02, estimate 95% CI = -0.05-0.09) and commercial goods ownership (importance = 0.26, estimate = 0.02, estimate 95% CI = -0.05-0.08) were weak predictors of centrality on the environmental overlap networks. Overall, men who were wealthier (house made of higher quality materials and more owned goods) tended to be more central to their environmental networks (Fig 2).

**Table 2. Study participant demographics and variable distribution across the three villages.**

|  | Village A | Village S | Village M |
|---|---|---|---|
|  | n (%) | n (%) | n (%) |
| Total | 563 | 408 | 249 |
| **Gender** |  |  |  |
| Man | 295 (52.4) | 240 (58.8) | 139 (55.8) |
| Woman | 268 (47.6) | 168 (41.2) | 110 (44.2) |
| **Age** (years) |  |  |  |
| 18-20 | 86 (15.3) | 59 (14.5) | 15 (6.0) |
| 21-30 | 170 (30.2) | 158 (38.7) | 51 (20.5) |
| 31-40 | 113 (20.1) | 78 (19.1) | 65 (26.1) |
| 41-50 | 86 (15.3) | 49 (12.0) | 44 (17.7) |
| >50 | 108 (19.2) | 64 (15.7) | 74 (29.7) |
| **BMI** (kg/m$^2$) |  |  |  |
| <18.5, Underweight | 78 (13.9) | 38 (9.3) | 36 (14.5) |
| 18.5-24.9, Healthy | 410 (72.8) | 318 (77.9) | 193 (77.5) |
| 25-29.9, Overweight | 69 (12.3) | 48 (11.8) | 17 (6.8) |
| ≥30, Obese | 6 (1.1) | 4 (1.0) | 3 (1.2) |
| **Education** |  |  |  |
| No formal schooling | 14 (2.5) | 16 (3.9) | 14 (5.6) |
| Primary | 238 (42.3) | 171 (41.9) | 150 (60.2) |
| Secondary | 235 (41.7) | 168 (41.2) | 62 (24.9) |
| Higher | 76 (13.5) | 53 (13.0) | 23 (9.2) |
| **Durable Goods*** |  |  |  |
| 0 | 146 (25.9) | 106 (26.0) | 67 (26.9) |
| 1 | 235 (41.7) | 133 (32.6) | 97 (39.0) |
| 2 | 121 (21.5) | 86 (21.1) | 47 (18.9) |
| >2 | 61 (10.8) | 83 (20.3) | 38 (15.3) |
| **Household Size**** |  |  |  |
| 1 | 40 (7.1) | 24 (5.9) | 23 (9.2) |
| 2-3 | 238 (42.3) | 183 (44.9) | 80 (32.1) |
| 4-5 | 201 (35.7) | 130 (31.9) | 86 (34.5) |
| >5 | 84 (14.9) | 71 (17.4) | 60 (24.1) |
| **House Materials#** |  |  |  |
| 0 | 95 (16.9) | 5 (1.2) | 36 (14.5) |
| 1 | 125 (22.2) | 81 (19.9) | 94 (37.8) |
| 2-4 | 271 (48.1) | 278 (68.1) | 94 (37.8) |
| >4 | 72 (12.8) | 44 (10.8) | 25 (10.0) |
| **Land Size** (ha) |  |  |  |
| <3 | 132 (23.4) | 151 (37.0) | 95 (38.2) |
| 3-8 | 289 (51.3) | 183 (44.9) | 112 (45.0) |
| >8 | 142 (25.2) | 74 (18.1) | 42 (16.9) |
| **Livestock$^\phi$** |  |  |  |
| 0 | 320 (56.8) | 237 (58.1) | 150 (60.2) |
| 1-3 | 201 (35.7) | 146 (35.8) | 73 (29.3) |
| >3 | 42 (7.5) | 25 (6.1) | 26 (10.4) |

*Durable Goods: Sum of self-reported goods owned by individuals, including cell phone, television, bicycle, refrigerator, motorcycle, computer, and generator.

**Household Size: Total members of a household.

#House Materials: Index calculated based on the material the house wall, roof, and floor are made of, see methods for more details. The higher the index, the better the building material quality.

$^\phi$Livestock: Sum of cows, goats, and pigs owned by households.

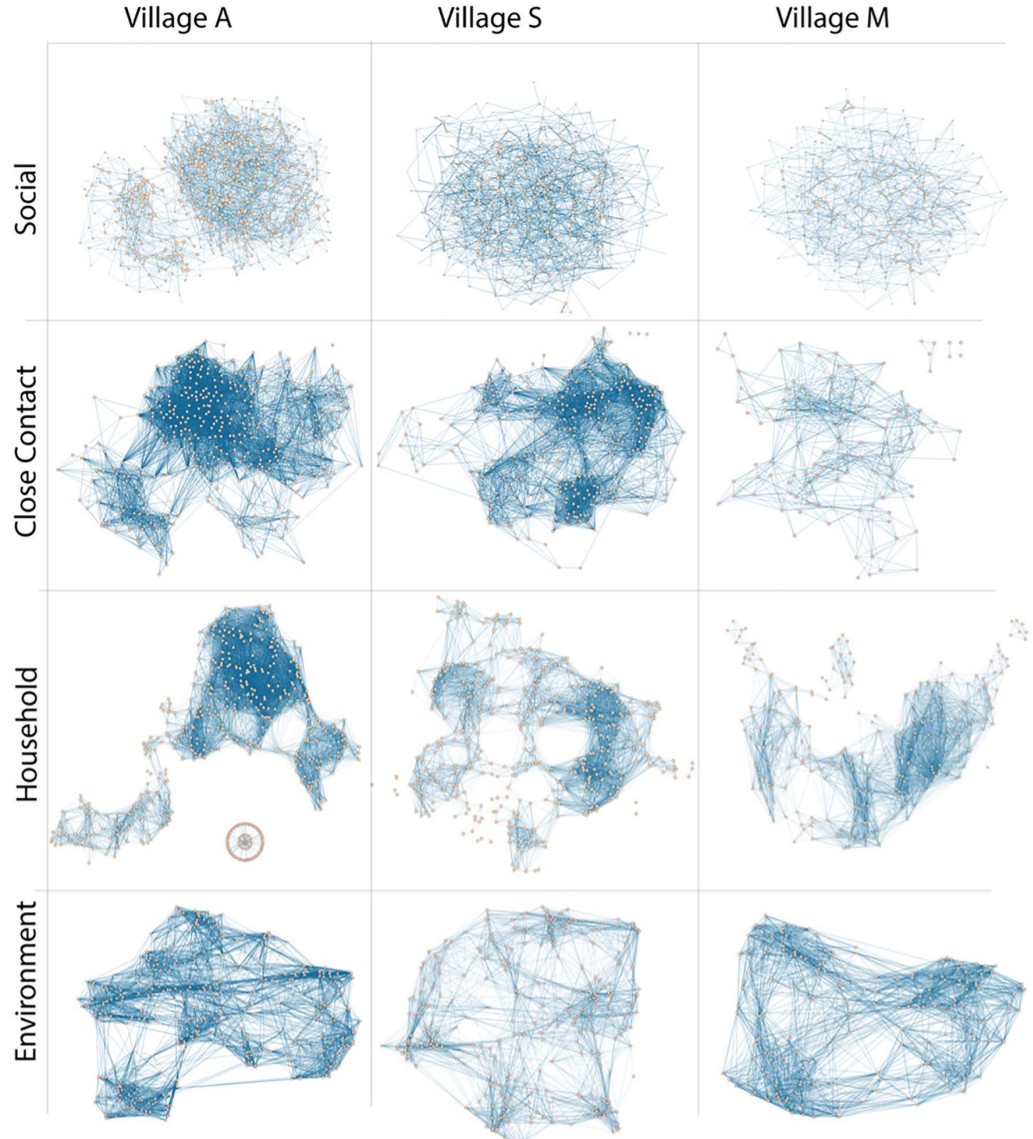

**Fig 1. Network visualizations for each village and network type.** Nodes represent individuals and edges represent their connections. Substructures that appear disconnected in these networks were based on applied thresholds that helped visualize connections; there were no completely disconnected network fragments in our study.

We conducted sensitivity analyses to assess whether sampling date impacted centrality estimates and the findings for other predictors, as might occur through snowball sampling. Gender and livestock in the social network were no longer important when sampling date was included in the model and people who were surveyed later in the sampling season tended to be less central to their networks (S3 Fig and S4 Fig). However, the estimated effects of the individual and household level predictors were generally consistent with the main analysis when including sampling date as a predictor (S3 Fig). We also minimized related selection biases by controlling for sampling season and village in the models.

Comparing the different centrality metrics separately (strength, PageRank, eigenvector, and betweenness; S5 Fig, S1 Table for the pooling analysis) showed that the metrics were positively correlated, but correlations varied by centrality

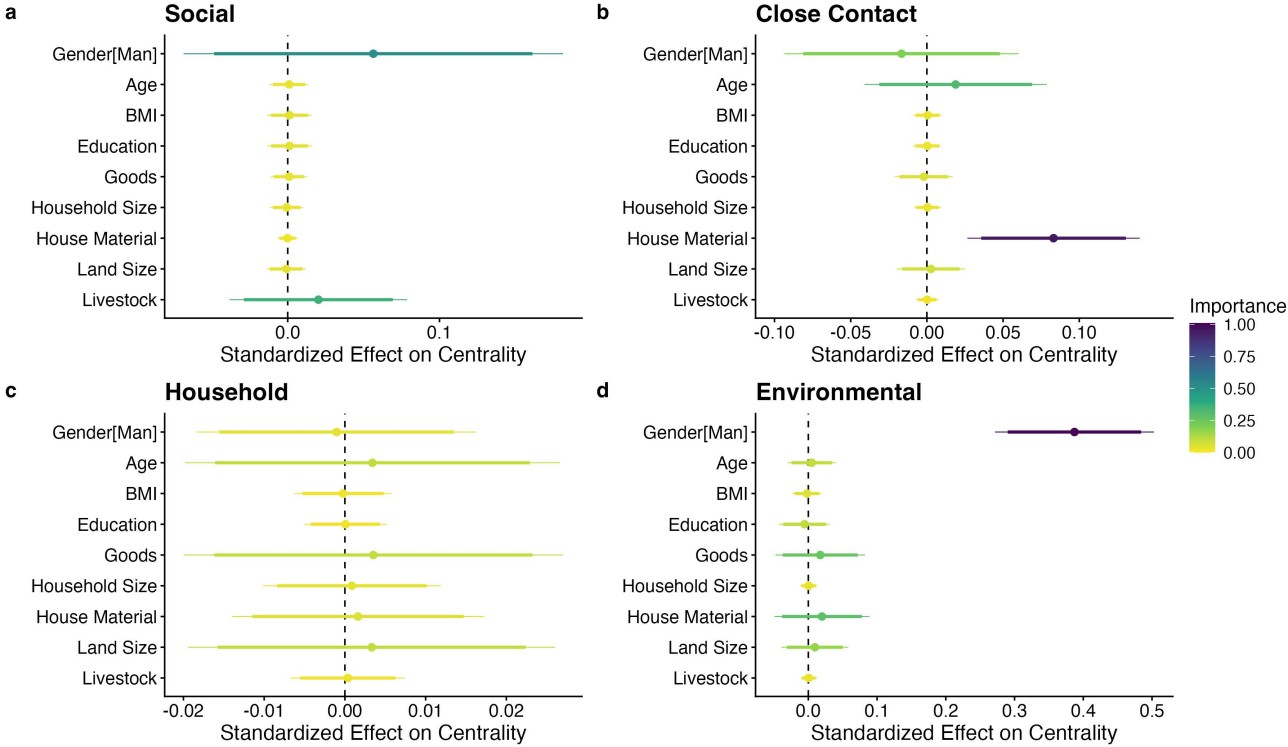

**Fig 2. Coefficient plots of the relationships between sociodemographic variables and centrality for each network type (a. Social Network, b. Close Contact Network, c. Household Network, and d. Environmental Network).** Each panel represents a different model. Points represent estimated effects; thick bars represent 90% confidence intervals and thin bars represent 95% confidence intervals; color represents variable importance. Models control for village, season, and centrality metric type (S7 Fig).

and across the four network types (S6 Fig). Betweenness centrality was the least correlated with other centrality metrics for the close contact, household, and environmental overlap networks. Village and sampling season tended to be the strongest predictors of centrality across network types, with individuals sampled in the two wet seasons tending to have lower centrality than individuals sampled in the dry season (S7 Fig). Notably, these are not individual- or household-level variables.

## Discussion

Building on prior research [14,19,23,46], we evaluated a comprehensive suite of sociodemographic and anthropometric variables as network centrality predictors among study participants who lived in three rural communities in the SAVA region of Madagascar. We evaluated centrality (strength, PageRank, eigenvector, and betweenness) across four network types (social, close contact, household, and environmental overlap). Notably, these four network types represent different transmission potential networks, capturing infectious diseases transmitted through direct contact and sharing of space (i.e., environmental overlap).

Social networks were assessed because they capture social relationships and the potential pathogen transmission through long-term direct contact and closely shared food, water, and space [47]. Identifying sub-populations that are most central to social networks can inform educational public health interventions by identifying and reducing high-risk interactions during infectious disease outbreaks. GPS-based close contact networks capture individual proximity and reflect direct contact-based transmission that does not depend on social interactions [48]. Like the social network, identifying

sub-populations that are most central to these networks can help identify and reduce high-risk interactions or inform vaccine deployment strategies during infectious disease outbreaks. Household networks capture household distances within a village and reflect potential infectious disease transmission via direct contact, shared environments around the house, and water [49]. Environmental networks capture GPS-derived environmental overlap outside of the village and reflect potential environmental infectious disease transmission based on shared land use [50]. Household and environmental networks are helpful to inform public health interventions inside and outside of village perimeters and enable comparison of transmission potential in the two settings.

We calculated four different centrality metrics because each captures slightly different aspects of node connections and therefore represent different pathways of potential infectious disease transmission. Strength (i.e., weighted degree centrality) accounts for frequency and duration of contact. For close contact infectious disease transmission, it is important to account not only for contact but also its intensity because longer contact increases the risk of transmission. Strength centrality has been applied in other infectious disease studies [2,38]. PageRank centrality mirrors possible movement of a pathogen or information through a network and is therefore a useful centrality measure when investigating disease transmission [40]. Eigenvector centrality (also known as power centrality) reflects the degree to which a node is connected to other highly connected nodes, and thus also captures connectivity of nodes, making it well suited for studies of infectious disease transmission [2,38]. Betweenness centrality quantifies the importance of a focal node functioning as a bridge between subgroups, therefore representing potential infectious disease transmission between clusters of other nodes. In our population and networks, the different centrality metrics were all positively correlated, but betweenness centrality was the least correlated compared to the other centrality metrics. Our results therefore suggest that, while PageRank, eigenvector, and strength centrality were similar, betweenness may capture a distinct nuance for health interventions.

Gender and the house material index (i.e., a measure of wealth with higher values indicating higher quality house materials) tended to be the most important sociodemographic predictors of centrality, although results had wide confidence intervals and varied by network type (Fig 2). Men were more central than women in the environmental overlap networks, and to some extent their social networks. Meaning, men were more likely to share geographic space outside of the village, reflecting their high social engagement and cooperative agricultural activities. Women tended to be more central than men in the close contact networks. However, the importance of gender in the close contact network was lower than the environmental overlap and social network, and the confidence intervals overlapped zero (Fig 2). These patterns are consistent with men spending more time outside of the village and women spending more time in the village.

These results align with other studies that also identified demographic variables like gender (or sex) to be related to centrality. Moore et al. [51], for example, found that female farmers were significantly less socially connected than male farmers in Madagascar. Other studies of rural communities determined that relationships among female kin are particularly strong and long-lasting, although the female relationships seem to be tied less to reputation (i.e., not linked to high positions or influence) than in males [52,53]. Mattison *et al.* [54], however, found greater differences in centrality between communities than between men and women when analyzing social network data on men's and women's friendship ties in matrilineal and patrilineal Mosuo communities in China. Therefore, while gendered centrality outcomes and the associated contact heterogeneities may be important, they are likely to differ across space, time, and culture.

Our results suggest that public health interventions, in settings like the rural Malagasy villages studied here, should consider gender-associated differences in contact patterns. Specifically, public health interventions (e.g., surveillance, vaccination, and diagnostic testing) should target men when addressing control of parasites transmitted through shared soil or water (i.e., hookworm and *Giardia*) or for zoonotic pathogens being transmitted through wildlife or livestock outside the homes and villages (i.e., avian influenza and Ebola). However, if pathogens are likely transmitted through close contact in household or village settings, for example through fecal-oral transmission (i.e., norovirus and shigella), then interventions should consider targeting women. Overall, these results highlight the importance of estimating predictors of centrality for more than one network type to capture the nuances of transmission potential.

Participants who lived in houses made of higher quality materials tended to be more central to their close contact networks. Therefore, in this rural agricultural setting, wealthier individuals were more likely to interact with others in their community. Focusing on individuals with these characteristics may help maximize the effectiveness of targeted interventions during outbreaks of infectious diseases that are transmitted through person-to-person contact, for example respiratory infections such as SARS-CoV-2. Our results align with other studies that identified socioeconomic predictors of contact networks. For example, Luo *et al*. [55] found that individual economic status was positively related to network centrality. The relationships between socio-economic status and network centrality are, however, context dependent; for example, in the United States, household income is negatively related to time spent socializing with others [56]. Furthermore, wealth indicators were not predictive of centrality in social and household networks and were only slightly important for the environmental overlap network.

While individual and household characteristics made limited contributions to predicting centrality, higher level variables were consistently influential. Individuals sampled in the two wet seasons tended to have lower centrality than individuals sampled in the dry season (S7 Fig). Weather could have driven differential movement patterns, especially for the GPS-based close contact and environmental networks. Seasonal differences in human movement have been observed in many other settings and have implications for disease dynamics [57,58]. However, the significant effects of sampling season in the temporally static household network (S7c Fig) point towards this effect being at least partially an artifact of snowball sampling, since wet seasons were sampled after the dry season. Also, varying village-dependent environmental factors and sizes could have led to different social patterns, movements, and settlement patterns. For example, individuals in Village A tended to be less central to their networks than individuals in Village M or S. This could be because Village A is composed of two proximate communities rather than one cohesive community. We advocate for future research across more villages, which would enable quantitative analysis of village level heterogeneities in human contact.

This study had several limitations. The GPS overlap measured in the close contact networks did not always indicate temporal overlap because not all study participants wore their GPS trackers at the same time [14]. Our analyses are based on self-reported data, which carry inherent limitations such as recall inaccuracies or reporting errors regarding possessions, land size, or other modeled variables. Further, while the centrality metrics applied in this study are relevant to networks in the context of infectious diseases transmitted between people, additional centrality metrics such as information centrality, as well as network community structure, could be used to measure different aspects of node importance in other contexts [59]. Similarly, investigating infectious diseases with other transmission modes – such as sexually transmitted diseases – will require different transmission potential networks.

There was a potential for selection bias in our snowball sampling recruitment design. There was no formal population census data available for our catchment area (i.e., the exact proportion of the overall population our study covered is unknown). However, the village chiefs provided estimated population sizes of adults and children in the villages (estimates for A, S, and M were 1900, 900, and, 2700 respectively) and our observations confirmed the range of age, gender, and socioeconomic status across the villages. We therefore expect that our recruitment captured a sizable portion of the adult population and sociodemographic variation within each village, and that the study cohort is therefore representative of the three villages at the time of data collection. We also implemented a sensitivity analysis, which revealed that findings related to individual and household-level variables were largely unaffected by the inclusion of sampling date, except for gender and livestock importance in the social network. We also minimized potential selection biases by controlling for village and sampling season in the models. Further, previous research has assessed the effect (i.e., bias) of non-random sampling and missing data in network analyses [29–31]. For networks similar to ours, a sample could have >70% missing with a strong bias toward missing peripheral nodes and still achieve a 0.9 correlation with any degree measure, while betweenness is more sensitive. Thus, a study would need a high under-sampling with extreme bias toward peripheral people to have a substantive effect on analyses.

Overall, gender and wealth were the strongest predictors of centrality in agricultural communities of rural northeast Madagascar, but the effect of these predictors was often uncertain, low, and variable across network types. Still, with

these and other proxies, it is possible to rank people according to their likely centralities [19,46]. Interventions may therefore consider using these data to monitor, treat, or vaccinate those with highest observed or predicted centrality to maximize the impact of intervention effort [19]. The effect of predictors differed across network types, with no sociode-mographic variables predicting centrality to household networks. Efforts to use centrality proxies must therefore carefully consider the transmission pathways and other characteristics of the infectious disease, along with the specific social and ecological contexts.

## Supporting information

**S1 Fig. Map of Madagascar (left) and the three villages covered by the study along the boundary of Marojejy National Park in the SAVA region of northeastern Madagascar (right, three blue dots).** Villages remain unnamed to protect the participants in the small villages (i.e., participants may be identified based on age, sex, and household charac-teristics due to the small village sizes). The Marojejy National Park shapefile was from UNEP-WCMC and IUCN (2025); the Madagascar country map shapefile was from Natural Earth (Natural Earth Data 2025) using the R package rnaturale-arth (Massicotte and South, 2025).
(DOCX)

**S2 Fig. Results of the principal component analysis (PCA) of wealth indicators: (a) Scree plot showing the pro-portion of variance explained by each principal component.** (b) PCA biplot of the first two principal components, with points representing individuals and arrows representing variable loadings for the six wealth indicators. (c) Pearson cor-relation heatmap of the six wealth indicators.
(DOCX)

**S3 Fig. Sensitivity analysis of including an index for sampling date in the models to account for the snowball sampling method.** Coefficient plots of the relationships between village, season, and centrality metric for each network type (a. Social Network, b. Close Contact Network, c. Household Network, and d. Environmental Network) when con-trolling for the sampling date index. Points represent estimated effects; thick bars represent 90% confidence intervals and thin bars represent 95% confidence intervals; color represents variable importance.
(DOCX)

**S4 Fig. Sensitivity analysis of including an index for sampling date in the models to account for the snowball sampling method.** Coefficient plots of the relationships between sampling date index, village, season, and centrality metric for each network type (a. Social Network, b. Close Contact Network, c. Household Network, and d. Environmental Network). Points represent estimated effects; thick bars represent 90% confidence intervals and thin bars represent 95% confidence intervals; color represents variable importance. The village base factor is Village A; the season base factor is Season 1; the centrality type base factor is betweenness.
(DOCX)

**S5 Fig. Coefficient plots of the relationships between socio-demographic variables and different centrality mea-sures for each network type (a. Social Network, b. Close Contact Network, c. Household Network, and d. Environ-mental Network).** Thick bars represent 95% confidence intervals and thin bars represent 90% confidence intervals. Each individual or household level variable is represented by a distinct color. Point shapes represent the centrality types.
(DOCX)

**S6 Fig. Correlations between centrality scores for the a. social, b. close contact, c. household, and d. environ-mental networks.** Color represents correlation value.
(DOCX)

**S7 Fig. Coefficient plots of the relationships between village, season, and centrality metric for each network type (a. Social Network, b. Close Contact Network, c. Household Network, and d. Environmental Network).** Points represent estimated effects; thick bars represent 90% confidence intervals and thin bars represent 95% confidence intervals; color represents variable importance. The village base factor is Village A; the season base factor is Season 1; the centrality type base factor is betweenness.
(DOCX)

**S1 Table. Centrality Metric Outputs for Metafor Analysis.** This table contains the numeric outputs and plotted confidence interval values for the estimated centrality measures in the model that considers the centrality metrics separately (i.e., metafor analysis). Results were generally consistent across models that compared the different centrality metrics separately (strength, PageRank, eigenvector, and betweenness).
(XLSX)

**S2 Table. Predictor Model Outputs.** This table contains the full model results and numeric outputs of the evaluated and plotted predictors in Fig 2.
(DOCX)

**S1 Text. Principal Components Analysis (PCA).**
(DOCX)

**S1 Checklist. Inclusivity in global research.**
(PDF)

## Acknowledgments

We thank the study participants and their families for their valuable time and participation. Without them, this study would not have been possible. We thank the Duke Lemur Center SAVA Conservation for valuable logistical support. We are also grateful for the Malagasy Medical Ethics Panel for permission to conduct the research. We thank Dr. Peter Mucha for valuable feedback on the manuscript content and applied methods. We are grateful to the Malagasy medical ethical review committee, Ministère de Santé Publique; the Mention Zoologie et Biodiversité Animale, Université d'Antananarivo; Madagascar National Parks; and the Ministère de l'Environnement et du Développement Durable for their support on the administrative procedures for research permits.

## Author contributions

**Conceptualization:** Camille M. M. DeSisto, Raquel A. Binder, James Moody, Charles L. Nunn.

**Data curation:** Camille M. M. DeSisto, Raquel A. Binder, Kayla Kauffman, Tyler M. Barrett, Michelle Pender, Voahangy Soarimalala, Jean Yves Rabezara, Prisca Rahary, James Moody.

**Formal analysis:** Camille M. M. DeSisto, Raquel A. Binder, Kayla Kauffman, James Moody.

**Funding acquisition:** Raquel A. Binder, James Moody, Charles L. Nunn.

**Investigation:** Camille M. M. DeSisto, Raquel A. Binder, Kayla Kauffman, Tyler M. Barrett, Randall A. Kramer, Voahangy Soarimalala, Jean Yves Rabezara, Prisca Rahary, James Moody, Charles L. Nunn.

**Methodology:** Camille M. M. DeSisto, Kayla Kauffman, Tyler M. Barrett, Prisca Rahary, James Moody, Charles L. Nunn.

**Project administration:** Michelle Pender, Voahangy Soarimalala, Jean Yves Rabezara, Prisca Rahary, Charles L. Nunn.

**Resources:** Kayla Kauffman, Michelle Pender, Randall A. Kramer, Voahangy Soarimalala, Jean Yves Rabezara, Charles L. Nunn.

**Supervision:** Voahangy Soarimalala, James Moody, Charles L. Nunn.

**Validation:** Camille M. M. DeSisto, Tyler M. Barrett, Michelle Pender, Voahangy Soarimalala, James Moody, Charles L. Nunn.

**Visualization:** Camille M. M. DeSisto, James Moody.

**Writing – original draft:** Camille M. M. DeSisto, Raquel A. Binder, Charles L. Nunn.

**Writing – review & editing:** Camille M. M. DeSisto, Raquel A. Binder, Kayla Kauffman, Tyler M. Barrett, Michelle Pender, Randall A. Kramer, Voahangy Soarimalala, Jean Yves Rabezara, Prisca Rahary, James Moody.

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
