## [Decision Letter · Decision Letter 0]

22 Aug 2025

PGPH-D-25-01742

Spreading Potential in Disease Relevant Networks: Predicting Centralities in Rural Northeast Madagascar

Dear Dr. Binder,

Thank you for submitting your manuscript to PLOS Global Public Health. After careful consideration, we feel that it has merit but does not fully meet PLOS Global Public Health’s publication criteria as it currently stands. Therefore, we invite you to submit a revised version of the manuscript that addresses the points raised during the review process.

The reviewers have noted several methodological and inferential issues, including the interpretation of predictors with limited evidence, possible selection bias caused by insufficient information on population coverage and sampling strategy, and a lack of details on GPS usage incentives and instructions. The significance scores, village-level variations such as household density, and the potential of other methods (e.g., wealth quintiles via PCA) to better represent socioeconomic effects require further explanation. Please address the specific comments on survey design, data collection, representativeness, and public health implications.

We look forward to receiving your revised manuscript.

Kind regards,

Giridhara Rathnaiah Babu, MBBS, MPH, PhD

Academic Editor

Journal Requirements:

2. Please provide separate figure files in .tif or .eps format.

3. **[** Our data contains GPS coordinates of the study participants that cannot be made fully available.]

We note that you have indicated that there are restrictions to data sharing for this study. For studies involving human research participant data or other sensitive data, we encourage authors to share de-identified or anonymized data. However, when data cannot be publicly shared for ethical reasons, we allow authors to make their data sets available upon request. For information on unacceptable data access restrictions, please see http://journals.plos.org/plosone/s/data-availability#loc-unacceptable-data-access-restrictions.

Please update your Data Availability statement in the submission form accordingly

Additional Editor Comments (if provided):

Reviewers' comments:

Reviewer's Responses to Questions

**Comments to the Author**

1. Does this manuscript meet PLOS Global Public Health’s publication criteria?

Reviewer #1: Yes

Reviewer #2: Yes

2. Has the statistical analysis been performed appropriately and rigorously?

Reviewer #1: I don't know

Reviewer #2: Yes

3. Have the authors made all data underlying the findings in their manuscript fully available (please refer to the Data Availability Statement at the start of the manuscript PDF file)?

Reviewer #1: Yes

Reviewer #2: No

4. Is the manuscript presented in an intelligible fashion and written in standard English?

Reviewer #1: Yes

Reviewer #2: Yes

Reviewer #1: This study has interesting public health applications when considering disease prevention activities, especially in a rural population.

Many of my comments request clarifying details to the study methods.

The methods section begins with details on data collection. I think the paper would benefit from additional details on the sampling strategy and the survey instrument. For example, how were individuals 18+ selected to be in the study? What proportion of the overall population does the study sample represent?

Additionally, details on data collection timing would be helpful. It’s noted that data collection happened over a nearly three-year period, but how much of that was active data collection time? These details would be especially useful for movement data collected with GPS trackers. It’s unclear for how long study participants recorded their movement, and how much data collection time varied across study participants and villages.

I think the reader would also benefit from more details on the survey. Was this part of a larger survey? Are there additional data that could be considered to determine network centrality (e.g. occupation)?

The survey collected data on durable goods ownership and household material construction, both of which are included in the model as a continuous index. Often with such scales, a score is constructed using principal components analysis and then individuals or households are broken into quintiles. I wonder if this would have an impact on results or be a more useful framing for understanding how wealth and centrality relate.

In Table 1, the BMI distribution and percentage of people attending higher education represent a higher proportion of the study sample than I would have expected. Additional information in the methods on how the sample was collected and some background information on the study area would help provide context for the reader on how representative this sample is. Also, a map to indicate how geographically proximate the three villages are could be helpful.

The discussion could include more reflection on how different network types and demographic predictors of centrality could be used to inform public health interventions.

Reviewer #2: This study investigates how sociodemographic and anthropometric characteristics predict individuals’ centrality in different contact networks. Incorporating GPS tracking and survey data in a large cohort of participant, the authors employed mixed-effect linear models and found gender and household material index (a proxy for wealth) to be often associated with higher centrality across networks.

This study addresses an important topic in network epidemiology: identifying likely “superspreaders” via low-cost but effective predictors. While the paper has potential, several methodology and interpretation concerns need to be addressed:

Major concerns:

1. While the study enrolled a large number of participants (n=1220), I was not able to gauge what proportion of the entire population was involved in this study without further information. The primary concern lies in the fact that the close contact and household networks tracked only participants in this study, but had no imputation or other means to account for those who did not participate. How is selection bias being addressed in the models?

2. Related to the first concern, was any incentive paid to the participants for them to wear the GPS tracker? What were the exact instructions provided to the participants about wearing the GPS device (e.g., wear it at all times vs. feel free to remove it as needed)?

3. Most of the predictors were not statistically significant in this study. For example, gender was the most important centrality predictor for the social network, but its confidence interval was wide and included 0. Does this suggest that even though gender and livestock ownership were important predictors, we cannot say much about the direction of their effects on centrality at a 95% confidence level? In general, I would be very careful about interpreting the results when the CI is wide and overlaps 0.

4. For the environmental network, gender had an importance score of 1. What does this mean, and how should I interpret this result statistically?

5. There is an imbalance in terms of household size and land size among the three villages. In particular, Village M had the highest proportion of households with more than 5 members, but the lowest proportion with large land sizes, which suggests a higher density of residents in this area. Does population density and/or other differences between the villages play a role in this analysis?

Minor concerns & typos:

1. The Results & Conclusions in the Abstract are too generic. I would recommend spelling out the key findings for each network type.

2. The authors noted that “village and sampling season tended to be the strongest predictors of centrality across network types” without providing more details about why this would have been the case. I suggest that the authors elaborate further and explain, for example, how seasonality affects the centrality in the analysis?

3. I found Supplementary Table 1 to be pretty helpful, in addition to the visualization in Figure 2. I suggest that the authors clean up and summarize the model results (with importance added as another column) and report the numbers.

4. On Page 13 Line 250, a decimal point was mistyped as a comma.

**Do you want your identity to be public for this peer review?** For information about this choice, including consent withdrawal, please see our Privacy Policy

Reviewer #1: No

Reviewer #2: No

---

## [Decision Letter · Decision Letter 1]

4 Nov 2025

PGPH-D-25-01742R1

Spreading Potential in Disease Relevant Networks: Predicting Centralities in Rural Northeast Madagascar

Dear Dr. Binder,

Thank you for submitting your manuscript to PLOS Global Public Health. After careful consideration, we feel that it has merit but does not fully meet PLOS Global Public Health’s publication criteria as it currently stands. Therefore, we invite you to submit a revised version of the manuscript that addresses the points raised during the review process.

The reviewer has identified concerns regarding the lack of pathogen-specific relevance, the discussion of sampling biases from snowball recruitment, insufficient details on network construction, unclear justification of centrality metrics, and limited discussion of village- and season-level variation. Thereby, please offer a clearer epidemiological framing, stronger contextualization, and cautious interpretation of weak effects in your revision.

We look forward to receiving your revised manuscript.

Kind regards,

Giridhara Rathnaiah Babu, MBBS, MPH, PhD

Academic Editor

Journal Requirements:

Additional Editor Comments:

The reviewer has identified concerns regarding the lack of pathogen-specific relevance, the discussion of sampling biases from snowball recruitment, insufficient details on network construction, unclear justification of centrality metrics, and limited discussion of village- and season-level variation. Thereby, please offer a clearer epidemiological framing, stronger contextualization, and cautious interpretation of weak effects in your revised manuscript.

Reviewers' comments:

Reviewer's Responses to Questions

**Comments to the Author**

Reviewer #2: All comments have been addressed

Reviewer #3: (No Response)

publication criteria?

Reviewer #2: Yes

Reviewer #3: Yes

3. Has the statistical analysis been performed appropriately and rigorously?

Reviewer #2: Yes

Reviewer #3: Yes

4. Have the authors made all data underlying the findings in their manuscript fully available (please refer to the Data Availability Statement at the start of the manuscript PDF file)?

Reviewer #2: No

Reviewer #3: Yes

5. Is the manuscript presented in an intelligible fashion and written in standard English?

Reviewer #2: Yes

Reviewer #3: Yes

Reviewer #2: All my comments have been addressed in the revision. The authors' approach to address and minimize selection bias inherent in the sample mechanism is reasonable, and the study results can add great value to the existing literature. I enjoyed reading the revised manuscript. Great work!

Reviewer #3: This manuscript explores how sociodemographic and anthropometric factors predict network centralities in four types of “transmission-potential” networks (social, close contact, household, and environmental overlap) in rural northeast Madagascar. The authors integrate survey and GPS-based data with network and mixed-effects modeling approaches to identify predictors of centrality that may correspond to higher disease transmission potential.

The study is methodologically ambitious. The integration of social, spatial, and epidemiological perspectives is commendable, and the data collection appears rigorous and ethically conducted. However, while the work is innovative, it requires stronger contextualization, a clearer epidemiological framing, and more cautious interpretation of statistically weak effects. Several methodological clarifications are also necessary before the manuscript can be accepted.

Major Comments

Epidemiological Relevance and Framing. The manuscript discusses “transmission-potential networks” abstractly but does not link the findings to specific pathogens or transmission modes relevant to Madagascar. Discussing about which pathogens could be relevant in this context, especially according to their transmission routes, is important to add.

Sampling and Representativeness. The snowball sampling approach is understandable for social network analysis, yet it introduces strong structural biases (especially under-representation of peripheral nodes). The authors should discuss more explicitly how this design, maybe thourgh simulations, may affect the observed centrality patterns and whether the resulting networks approximate whole-community structures.

Network Construction Details. While Table 1 summarizes the types of networks, key methodological details remain vague—particularly the criteria for edge weighting and the treatment of non-synchronous GPS tracking. A clear schematic or expanded supplementary table explaining how proximity thresholds and environmental overlaps were defined would strengthen reproducibility.

Choice and Interpretation of Centrality Measures. The paper uses multiple centrality metrics (strength, PageRank, eigenvector, betweenness) without fully explaining their epidemiological meaning. Why, for instance, is PageRank used for undirected networks? A short conceptual justification of each metric’s relevance to transmission would help readers interpret the results.

Village- and Season-Level Variation. Village and season effects appear important, yet the discussion remains brief. Since these represent key ecological and social contexts, the authors should explore (even qualitatively) how these factors might shape contact patterns and centralities differently across locations.

Minor Comments

- The abstract could be streamlined for clarity and consistency in style (avoid redundancy such as “few individuals are responsible for substantially more secondary infections…”).

- Figures are informative but could benefit from clearer legends and consistent font size.

- A concise summary table highlighting significant predictors by network type would help readers synthesize the results.

**Do you want your identity to be public for this peer review?** For information about this choice, including consent withdrawal, please see our Privacy Policy

Reviewer #2: No

Reviewer #3: No

---

## [Decision Letter · Decision Letter 2]

5 Jan 2026

Spreading Potential in Disease Relevant Networks: Predicting Centralities in Rural Northeast Madagascar

PGPH-D-25-01742R2

Dear Dr. Binder,

We are pleased to inform you that your manuscript 'Spreading Potential in Disease Relevant Networks: Predicting Centralities in Rural Northeast Madagascar' has been provisionally accepted for publication in PLOS Global Public Health.

Best regards,

Giridhara Rathnaiah Babu, MBBS, MPH, PhD

Academic Editor

Reviewer Comments (if any, and for reference):

Reviewer's Responses to Questions

**Comments to the Author**

Reviewer #3: All comments have been addressed

publication criteria?

Reviewer #3: Yes

3. Has the statistical analysis been performed appropriately and rigorously?

Reviewer #3: Yes

4. Have the authors made all data underlying the findings in their manuscript fully available (please refer to the Data Availability Statement at the start of the manuscript PDF file)?

Reviewer #3: Yes

5. Is the manuscript presented in an intelligible fashion and written in standard English?

Reviewer #3: Yes

Reviewer #3: (No Response)

**Do you want your identity to be public for this peer review?** For information about this choice, including consent withdrawal, please see our Privacy Policy

Reviewer #3: No
